# Deep Physiological Model for Blood Glucose Prediction in T1DM Patients

**DOI:** 10.3390/s20143896

**Published:** 2020-07-13

**Authors:** Mario Munoz-Organero

**Affiliations:** Telematic Engineering Department and UC3M-BS Institute of Financial Big Data, Universidad Carlos III de Madrid, Leganes, 28911 Madrid, Spain; munozm@it.uc3m.es

**Keywords:** blood glucose prediction, type 1 diabetes mellitus, deep machine learning, physiological models

## Abstract

Accurate estimations for the near future levels of blood glucose are crucial for Type 1 Diabetes Mellitus (T1DM) patients in order to be able to react on time and avoid hypo and hyper-glycemic episodes. Accurate predictions for blood glucose are the base for control algorithms in glucose regulating systems such as the artificial pancreas. Numerous research studies have already been conducted in order to provide predictions for blood glucose levels with particularities in the input signals and underlying models used. These models can be categorized into two major families: those based on tuning glucose physiological-metabolic models and those based on learning glucose evolution patterns based on machine learning techniques. This paper reviews the state of the art in blood glucose predictions for T1DM patients and proposes, implements, validates and compares a new hybrid model that decomposes a deep machine learning model in order to mimic the metabolic behavior of physiological blood glucose methods. The differential equations for carbohydrate and insulin absorption in physiological models are modeled using a Recurrent Neural Network (RNN) implemented using Long Short-Term Memory (LSTM) cells. The results show Root Mean Square Error (RMSE) values under 5 mg/dL for simulated patients and under 10 mg/dL for real patients.

## 1. Introduction

Type 1 Diabetes Mellitus (T1DM) affects the capacity of the pancreas to produce insulin and affects the Blood Glucose (BG) regulation mechanisms in the body. Patients suffering T1DM have to monitor their BG levels, insulin injections and food intake in order to continuously adjust them. Very high or low values of BG levels can cause different inconveniencies and damages to the human body. Mechanisms able to anticipate them could increase the quality and even save lives for those suffering T1DM.

Artificial BG level regulation systems work based on three major parts: data gathering, control algorithms and actuation mechanisms. Data gathering obtains BG related measurements either automatically from different body worn sensors or manually introduced readings from the patient. Continuous Glucose Monitoring (CGM) devices are commonly used by T1DM patients in order to get readings every few minutes. CGM devices have been commonly incorporated into clinical research studies [1]. Other sensors such as heart rate or acceleration measurement units could also be used in order to get additional data that may have an impact on upcoming values of BG levels. Control algorithms can be categorized into physiological–mathematical and machine learning models. Actuation mechanisms can be divided into open-loop and closed-loop systems depending on the agent responsible to act in order to adjust the insulin injections [2]. Providing models for accurate prediction of glucose levels in T1DM patients is critical both for their glycemic control and for the development of closed-loop systems [3].

This paper focuses on the use of the most common data sources/signals used in previous research studies for BG level estimation over prediction horizons of 30 to 60 min; including current and past BG measurements from CGM devices, fast and slow acting insulin injections and food intake. A horizon of 30 to 60 min of accurate predictions will allow the patient to modify insulin or meal intakes with enough time for the insulin and carbohydrate absorption in order to prevent adverse glycemic events. Based on this input information, a new physiological inspired model implemented using deep learning components which are trainable and adjustable to each particular user is developed and validated. The proposal outperforms previous models which use generic machine learning models in order to learn patterns from BG signals. The proposed model is trained and validated both on simulated data using the AIDA diabetes software simulation program [4,5] and with real patient data from the D1NAMO open dataset [6].

The paper is organized into the following sections. Section 1, this section introduces the objectives of the study. Section 2 summarizes the previous results from related studies and justifies the gap that the current paper covers. The proposed physiologically inspired machine learning model is described in Section 3. Section 4 is dedicated to describing the datasets used in order to validate the results. Section 5 captures the major results when applying the proposed model to both simulated and real data. Finally, Section 6 draws some conclusions and presents the future work.

## 2. Related Work

Machine learning techniques and methods have been widely used in order to predict and diagnose diabetes, predict health complications caused by the evolution of certain diabetes related symptoms, assess the genetic background and the environmental factors and even to provide support for health care management [7]. Zou et al. [8] used several classification methods based on decision trees, Random Forest and Neural Networks in order to classify participants into diabetic or healthy individuals showing accuracies of around 0.8 for two different datasets. Ashiquzzaman et al. [9] were able to improve the classification accuracy to 0.88 by adding dropout layers to the machine learning model. The authors in [10] provided a systematic updated review of the state of the art in machine learning for diabetes care in four main areas: automated retinal screening, clinical decision support, predictive population risk stratification, and patient self-management tools.

For T1DM diagnosed patients, the use of Continuous Glucose Monitoring (CGM) devices in combination with machine learning techniques has been widely used for predicting near future glycemic events. The research study in [11] used a Deep Believe Network (DBN) model and Electrocardiogram (ECG) signal to detect the natural occurrence of nocturnal hypoglycemia, using 15 children with T1DM who were monitored for 10 h overnight at the Princess Margaret Hospital for Children in Perth, Western Australia. Bertachi et al. [12] also investigated the feasibility of a machine-learning-based prediction model to anticipate Nocturnal Hypoglycemia (NH) in T1DM patients, using Continuous Glucose Monitoring (CGM) devices and physical activity trackers under free-living conditions at home. By using supervised machine learning algorithms, individualized prediction models were generated using a Multilayer Perceptron (MLP) and a Support Vector Machine (SVM), showing that more than 70% of the NH may be avoided using the proposed methodology. 

Machine learning techniques have also been applied to estimate upcoming values of BG levels. Accurate estimations for prediction horizons of 30 to 60 min will allow a T1DM patient to take appropriate actions in advance in order to avoid hypo and hyperglycemic episodes that will have a negative impact on the patient’s health. Accurate predictions are also the basis for the implementation of the artificial pancreas that makes the life for T1DM patients easier and more convenient. Pappada et al. [13] used a Neural Network-based model to predict blood glucose levels. The model incorporated food intake, insulin and exercise data, which was manually recorded in order to increase the precision of the BG level estimation. The algorithm worked well for predicting hyperglycemic episodes (with accuracies around 0.95 for a 60 min prediction horizon), but failed to predict hypoglycemic episodes. The authors in [14] described a model for blood glucose estimation/prediction that uses a Kalman filter in order to estimate hidden values in the model and implements a Support Vector Regression (SVR) algorithm to estimate future values based on current and past levels of carbohydrates, insulin and BG levels. The results are comparable to those manually predicted by a doctor. The authors in [15] also used an SVR model that, based only on Continuous Glucose Monitoring (CGM) data, tries to predict blood glucose levels independently of other factors, improving the results from other similar previous studies by adding Differential Evolution (DE) algorithms over data from 12 patients using CGM devices. The obtained average of the Root Mean Square Error (RMSE) was 10.78 and 12.95 mg/dL for Prediction Horizons (PHs), respectively, equal to 30 and 60 min. Ali et al. [16] proposed an improved method based on Artificial Neural Networks (ANNs) for the blood glucose level prediction of Type 1 Diabetes (T1D) using only CGM data as inputs validated on real CGM data of 13 patients, achieving RMSE values of 7.45 mg/dL and 9.03 mg/dL for Prediction Horizons (PHs), respectively, for 30 min and 60 minutes. The authors in [17] proposed a meta-learning approach based on the idea of using regularized learning algorithms in predicting blood glucose. Meta-learning approaches are designed to be portable from patient to patient while outperforming other algorithms in terms of clinical accuracy. This feature opens the way for using them in diabetes smartphone applications.

In order to improve the results for BG level estimation, several deep learning-based models have been proposed based on CGM signals, complemented sometimes with other wearable sensor data and manual recordings. The study in [18] investigated methods for deep multi-output blood glucose forecasting showing that the results using deep learning methods outperformed previous shallow learning alternatives. Mhaskar et al. [19] also proposed a deep learning approach to BG level estimation based on the previous BG levels but using a pre-clustering mechanism to train specific models for hypo, eu and hyperglycemic segments. The authors also demonstrated that deep learning methods can outperform shallow networks. The research study in [20] also presented an approach for predicting blood glucose levels for diabetics up to one hour into the future based on deep learning methods. The authors used a Recurrent Neural Network (RNN) based on Long Short-Term Memory (LSTM) cells trained in an end-to-end fashion, requiring nothing but the glucose level history for the patient. The method was validated using “The Ohio T1DM Dataset for Blood Glucose Level Prediction” [21], achieving RMSE values of 20.1 mg/dL for a 30 min prediction horizon and 33.2 mg/dL for a 60 min prediction horizon. Sun et al. [22] also used an LSTM model based on the BG signal to predict upcoming values for BG levels and compared results with Auto-Regressive Integrated Moving Average (ARIMA) and SVR models. The results outperformed previous methods, achieving RMSE values of 21.7 mg/dL and 36.9 mg/dL for prediction horizons of 30 and 60 min, respectively. A similar study using an LSTM RNN to predict upcoming values for BG levels can be found in [23]. The mean value of the RMSE of the model was 12.38 mg/dL based on data from 10 children and only used previous BG levels to estimate upcoming values. The authors in [24] tried to learn the chaotic properties in the glucose signal obtained from CGM systems using a model based on Echo State Network’s (ESNs) and achieved RMSE values of 13.57 mg/dL for a 30 min prediction horizon when implementing subject specific variants to the model. The authors in [25] proposed an approach based on Recurrent Neural Network’s (RNNs) trained in an end-to-end fashion, using the blood glucose signal, and were able to provide an estimate of the certainty in the predictions by training the recurrent neural network to parameterize a univariate Gaussian distribution over the output.

Although the main information for BG level predictions used in previous research studies is based on the current and recent past values from the CGM device measurements, adding data from other sources and wearable sensors that measure variables affecting the metabolic process could lead to optimized results [26]. The authors in [26] carried out a literature review regarding modeling options and strategies of machine learning focusing on the prediction of BG dynamics in type 1 diabetes. The authors recognized that due to the complexity of BG dynamics, it remains difficult to achieve a universal model that produces an accurate prediction in every circumstance (i.e., hypo/eu/hyperglycemia events) and that models adding information about food intake, insulin injections, physical exercise and mental health related parameters, such as stress levels, could improve BG level prediction results. Hayeri in [27] added heart rate, step-count and insulin information to the BG signal in order to improve results in the estimation of the next levels for BG. The proposed algorithm was applied to nine children with T1DM and the model was able to predict the user’s future glucose values with a 93% accuracy rate for 60 min ahead of time. Zhu et al. in [28] used CGM data together with the insulin values and carbohydrate intake estimations in the dataset in [21] and a deep learning model to achieve an RMSE value of 21.7 mg/dL on a 30 min prediction horizon. The research study in [29] used both CGM and insulin data and a deep learning model based on RNN with LSTM cells to predict the levels of BG in the next 30 min. The study achieved an RMSE value of 7.55 mg/dL and anticipated the occurrence of 97.79% of hyperglycemia events (glucose > 180 mg/dL), and 90.87% of hypoglycemia events (glucose < 70 mg/dL). Li et al. [30] also used insulin and carbohydrate intake information together with the BG signal from a CGM device and a deep learning model based on the combination of a Convolutional Neural Network (CNN) for automatic feature extraction and then an LSTM RNN for time series prediction in order to estimate the BG levels with a 30 and 60 min prediction horizon. The authors obtained RMSE values of 9.38 mg/dL for 30 min predictions and RMSE values of 18.87 for 60 min predictions. According to the results in [26], Hobbs et al. [31] designed a dynamic glucose prediction model that included both heart rate measurements and variables representing the carbohydrate consumption and insulin boluses in order to improve results for physically active adolescents. The authors achieved an RMSE value of 26.33 mg/dL for a 30 min prediction horizon which slightly improved the prediction results for the same dataset without including the heart rate information (RMSE value of 28.64 mg/dL for a 30 min prediction horizon).

The BG level prediction results from the machine learning models have also been compared with results provided by the mathematic metabolic models adjusted to fit each patient. Mirshekarian el al. in [32] implemented a deep learning model to fit BG signals based on RNN with LSTM cells and compared the achieved results with those achieved fitting a physiological model to the data. The machine learning model slightly outperformed the physiological model achieving RMSE values of 21.4 mg/dL and 38.0 mg/dL for prediction horizons of 30 and 60 min, respectively. The authors in [14] described a solution that used a generic physiological model of blood glucose dynamics to generate informative features for a Support Vector Regression (SVR) model that was trained on patient-specific data. The model outperformed diabetes experts at predicting blood glucose levels. The authors in [33] used the physiological models described in [34,35] to generate a 54-dimensional feature space. Then a deep learning model based on an RNN with LSTM cells was used based on the computed features. The authors recognized that one major drawback of physiological models is the requirement for prior knowledge to adjust the physiological parameters. The authors achieved an RMSE value in the best scenario of 14.04 mg/dL for a 60 min prediction horizon. The authors in [12] used the insulin and carbohydrate absorption models in [34,35] to try to estimate hypoglycemic events overnight for TD1M children. A review comparing physiological models partially or totally replaced by machine learning techniques can be found in [36].

In this paper, a new mechanism inspired by metabolic models for glucose dynamics [14,32,33] and trainable on a per-patient-basis is proposed. The differential equations for carbohydrate and insulin absorption are modeled using a Recurrent Neural Network (RNN) implemented using Long Short-Term Memory (LSTM) cells.

## 3. Proposed Model

The variations over time of blood glucose levels depend among other factors of current blood glucose levels, carbohydrate intake and insulin injections (according to the specific absorption rates from different insulin types). The glucose metabolic processes can be modelled using a set of differential equations that have been previously proposed in research studies such as [14,32,33]. Considering the carbohydrate intake, fast and slow acting insulin boluses and past blood glucose levels as inputs, the plasma glucose, insulin and carbohydrate levels depend on digestion, absorption, insulin dependent and independent utilization, renal clearance and endogenous liver production processes [14]. For the carbohydrate and insulin inputs, a Recurrent Neural Network (RNN) will be able to learn the digestion and absorption processes when trained together with the data from current values and the memory from the past. Combining the output signals from the absorption and digestion processes together with the blood glucose signal into a second learning layer, the plasma blood glucose level variations could be estimated.

The proposed method is captured in Figure 1. A specific Long Short-Term Memory based Recurrent Neural Network (LSTM RNN) is used to learn the carbohydrate digestion and insulin absorption processes from each input signal. The individual effect for each digestion and absorption process after the LSTM RNN layer is combined in order to assess the blood glucose variations for the next Continuous Glucose Monitor (CGM) reading. According to the mathematical metabolic models in [14,32,33], there are some BG variations influenced by current BG levels such as those induced by renal clearance, insulin independent BG utilization and endogenous liver production. The model in Figure 1 applies a different LSTM RNN network to learn the time patterns from the BG levels and combines the output of such RNN with the combined output from the processed insulin and carbohydrate signals in order to estimate the BG variation for the next CGM reading.

The question marks in the different layers in the model represent the number of time samples fed into the model. In order to take into account the influence over time for all the input signals, a time span of 9 h has been considered for the results presented in this paper. The simulated data used in this paper will produce data samples each 15 min, so in order to generate a 9 h data window a total of 36 samples are required. Different time spans could be used depending on the absorption curves for each of the inputs (in particular the type of insulin used).

The model will learn the Blood Glucose (BG) dynamics estimating the variation for the next expected CGM measure as captured from the differential equations in the metabolic models [14,32,33]. Different prediction horizons are calculated after the model is trained by executing the model in prediction mode as many times as needed in order to cover the required time span. The estimated BG variations are used to compute the estimated next values so that the model can be used to predict future samples using previously assessed data in a recurrent way. Since the model only predicts the evolution of the BG levels, the input values for the insulin and food intake required to recurrently run the model in prediction mode are set to 0. This will allow the model to unfold predictions based on the metabolic dynamics when no external inputs are added. Since future values for meals and insulin injections are not fed into the model, the model will generate an estimate about what will happen to the glucose signal if no external action is taken by the patient. Therefore, the model could be used to warn the user in advance about negative episodes if no action is taken and recommend particular actions to avoid such episodes.

## 4. Description of the Datasets

Two different datasets have been used in order to validate the proposed model: a dataset generated using a diabetes simulator in order to have the power to generate as many data samples as required and a second dataset with real data (containing a more limited set of data but in a real use case scenario). Both datasets contain information from meals, insulin boluses (slow and fast acting insulin) and CGM readings.

The first dataset has been generated using the AIDA diabetes simulator [4,5] which is intended for simulating the effects on the blood glucose profile of changes in insulin and diet for a typical insulin-dependent (type 1) diabetic patient. The simulator includes 40 different patient models with different parameters controlling the metabolic model that is used in order to generate BG levels for different food intake and insulin injection patterns. The AIDA diabetes simulator can be downloaded as a freeware tool or it can be used online. The simulator uses a 15 min sample rate for simulated BG levels.

The second dataset, the D1NAMO dataset [6], contains data for nine patients with type 1 diabetes. The data acquisition was made in real life conditions using a Zephyr BioHarness 3 wearable device (https://www.zephyranywhere.com/). Apart from insulin boluses, the dataset consists of ECG, breathing, and accelerometer signals, as well as glucose measurements and annotated food pictures. The pictures have been annotated by a nutritionist in order to estimate the number of calories taken by each participant. The information from the ECG, breathing and accelerometer have not been used in the current study since they are not present in the AIDA diabetes simulator in order to be able to use the same model for both datasets. The CGM readings provide BG levels every 5 min. There is around 4 days of information for each participant. The major limitation in the dataset (apart from the size of samples recorded) is that not all the meals taken by each participant are recorded in the dataset and the exact times in which the meals were taken are not available either. However, using the time of capture metadata for each picture for each meal, an estimate about when the recorded meals were taken has been generated.

## 5. Results

The model described in Section 3 has been trained using both datasets as described in Section 4. The model in Section 3 has been implemented in Python using Keras (https://keras.io/) and Tensorflow (https://www.tensorflow.org/) libraries. The Python code is captured in Appendix A.

This section captures the main results and compares them with results from previous related research studies.

### 5.1. Simulated Scenario

The AIDA diabetes simulator [4,5] has been used to generate 10 days of data for the different models implemented by the tool. Two different training and validation methods have been used to assess the quality of the model. The first validation method performs a 70–30% random split of the entire dataset. The differential equations in the mathematical metabolic blood glucose dynamics models define user dependent parameters. A user centered approach has therefore been used in order to adapt the proposed method in Section 3 to the metabolic functions of each participant. The second validation method uses 8 days of generated data for training and 2 different days for validation. Each day has been generated with different meal and insulin injection data so that each day has different BG signals. The idea with this second validation method is to assess the quality of the model in learning the metabolic processes so that it can generalize to new food and insulin scenarios.

#### 5.1.1. Validation Based on a 70–30% Data Split

The simulated data for each patient model in the AIDA simulator has been randomly split into a 70% training set, and the remaining 30% has been used for validation. Each training sample contains a 9 h segment of data for each of the four input signals (CGM readings, food intake, fast and slow insulin injections, 36 samples each). The model predicts the variations for the GCM signal in order to assess the next sample (according to the differential equations defining the metabolic model in [14,32,33]). Once the model is trained, the 30% validation samples are used to assess/estimate/predict the next Blood Glucose (BG) level for each 36-sample window. Then, the oldest sample in the 36-sample window is removed and the predicted values are added in order to estimate the values for the next window in time. The prediction process is repeated until the prediction horizon is reached. In order to compare results with previous related studies, two different horizons have been selected since they are normally present in previous research: 30 and 60 min horizons.

Previous research studies use different metrics in order to assess the quality of the proposed blood glucose models. Among them, the most widely used figure is the Root Mean Square Error (RMSE) defined by the following equation:(1)$$RMSE=\sqrt{\frac{\sum_{N}{\left(G_{m}-G_{e}\right)}^{2}}{N}}$$
where *G_m_* represents the measured BG level and *G_e_* is used for the estimated value for BG level at the same future instant of time.

Although RMSE has been widely used in previous publications and is therefore a convenient metric for comparing results among them, the Clarke Error Grid Analysis (EGA) is preferred in terms of clinical accuracy of blood glucose estimates [37]. The Clarke Error Grid captures the differences between the estimations for blood glucose levels over the selected horizon and the real measurements for the same instants of time. The Clarke Error Grid divides the bidimensional space into five different zones (A to E), each of them showing different implications in terms of blood glucose management. Zone A includes the predicted values that differ from the real values no more than 20%, or the values in the hypoglycemic range (<70 mg/dL). The points in this zone are considered clinically accurate. Points in Zone B capture values in which the difference between the prediction and real measurements are bigger than 20% but would lead to benign or no treatment scenarios. This zone is clinically acceptable. Zone C leads to estimations that imply overcorrecting behaviors based on the prediction errors for blood glucose levels. Points in Zone D represent a failure to detect and treat deviations in blood glucose levels in which the actual levels are outside of the acceptable levels while the predictions fall within the acceptable range. Finally, Zone E captures points in which predicted values are opposite to real/measured blood glucose levels, and therefore, the treatment would go in the opposite direction to what is recommended. In this paper, RMSE values will be used to compare results with previous studies and the Clarke Error Grid will also be used to assess the severity or the errors in the proposed method in clinical terms.

The model in Section 3 has different parameters that must be adjusted in order to better capture the dynamics behind the input signals. The major parameter of the proposed model consists of the number of memory units inside each LSTM cell. A low number of memory units will make the model unable to learn all the patterns in the signals while a big value could be prone to overfitting. Table 1 shows the results for the RMSE figures for three different values for the number of memory units inside each LSTM cell. Predictions over 30 and 60 min horizons are given. The optimum value for both prediction horizons is when the number of memory units is 10. This will be the value selected for the rest of the scenarios in this section. The influence is more notoriously noticed for longer term predictions.

Figure 2 and Figure 3 show the result for the estimated blood glucose levels for both 30 and 60 min horizons. The results for a 60 min horizon are a bit worse and bigger differences can be visually assessed per the relative maxima and minima values, however, both cases generate a low error RMSE figure.

The results can be observed in more detail by zooming out some particular days. Figure 4 shows the predicted vs. the real measured values for BG levels for two entire days. The model is able to better predict the raising BG level segments (normally caused after meal intakes) and, slightly worse, the falling segments (induced by insulin and fasting periods). Both images show similar results. The maximum and minimum instants (inflexion points) are estimated with low time lag figures.

Figure 5 captures the results for the Clarke Error Grid for a simulated participant for 30 and 60 min prediction horizons. The majority of the estimations fall in Zone A (clinically accurate zone) for both cases. In the 30 min horizon prediction, from the 739 points in the validation set, 738 fall in Zone A and only one in zone D. In the 60 min horizon case, 729 fall in Zone A, five in Zone B and only one in Zone D. Zone B is clinically acceptable while Zone D represents a failure to detect a hypo or hyperglycemic point in advance.

In order to assess how the model is able to learn the underlying metabolic processes, Figure 6 shows the predicted BG level variations on a given day together with the values of the carbohydrate intake and insulin boluses. The food intake increases the BG levels while insulin has the opposite effect. The bigger the input values, the bigger the variations expected.

#### 5.1.2. Validation Based on 8 Days for Training 2 Different days for Validation Data Split

A second validation method is used in this section. For each participant, 10 days of simulated data has been generated. The first 8 days are used for training the model in Section 3 while the information from the last 2 days is used for validation. The times, hours and amounts for the input signals (food and insulin) are different for all the generated days so that the time series in the validation period are therefore different from those seen by the model from the training data. The objective is to assess if the model can learn the underlying metabolic processes and generalize to new data for the same user.

A similar analysis in order to assess the optimal value for the number of memory units in each LSTM cell is shown in Table 2. The results are similar to those in Table 1, and an optimal value is obtained for 10 units. The RMSE values are a bit better in this case.

Figure 7 shows the predictions over a 60 min horizon for a particular day in the validation set. The graphical results show that the predictions follow significantly well the shape of the real predicted signal. In some cases, the predicted signal is able to anticipate the maxima and minima values with a bit more than 60 min.

### 5.2. Scenario with Real Data

For validating the model presented in Section 3 with real data, the D1NAMO dataset has been used [6]. The dataset contains nine T1DM patients wearing a CGM device providing readings every 5 min. The dataset also contains information for meals and insulin injections. The data for each participant is around 4 days and some of the meals are not recorded which can cause convergence problems in some cases for the model, especially if the number of memory cells were increased (overfitting problems). In order to avoid misleading the training of the model when including segments with missing data, data segments with a significant BG level increase likely to be caused by a meal intake that is not recorded are not taken into account. For these segments, the patient records the insulin boluses but not the related meal information. Moreover, data segments for which there are events in the meal or insulin signals in the prediction window period are not taken into account when validating the algorithm since these events modify the internal model dynamics as proposed in [38] and can not be predicted based on the information provided as input to the model (the authors in [38] used a similar approach).

In order to compare the results with simulated data, Table 3 captures the RMSE values for a similar model configuration (using 10 memory units per LSTM cell) using a 70% training and 30% validation split.

Figure 8 shows the prediction results for a participant in one of the days. The model adjusts both the hypoglycemic episode during the night and the hyperglycemic behavior during the day. The sensor used shows saturation around 400 mg/dL which represents a measurement error in this case. The prediction model tries to assess/predict values higher than 400 mg/dL at the beginning of the saturation window (when the model is fed with accurate data) but adjusts the predictions to the saturated samples for the final part of the glucose peak when the information used for estimating future values is based on saturated data samples.

The Clarke Error Grid for a real participant is shown in Figure 9. In this case, the prediction errors captured in Table 3 are bigger than the simulated case and the points outside Zone A in the Clarke Error Grid will be bigger. For the 30-min horizon, 868 points fall in Zone A, 205 in Zone B, 8 in Zone C, 13 in Zone D and there are no points in Zone E. The majority of the points fall therefore in zones A and B which are considered clinically accurate or acceptable. The 8 points in Zone C may lead to overreaction and the 13 points in Zone D will imply a failure to detect a glycemic episode 30 min in advance. For the 60 min horizon, 582 points are in Zone A, 367 in Zone B, 77 in Zone C, 57 in Zone D and 5 in Zone E. The majority of the points continue to fall in zones A and B but the number of error in zones C, D and E increase.

A final experiment has been done in order to assess the transferability of the model trained for one participant and applied to estimate the data for a different participant. The physiological glucose dynamics models in [14,32,33] have several metabolic parameters that have to be adjusted for each participant. In our case, when the model is trained for a participant, the training of the model will adjust its internal weights to predict the glucose variations for this participant. The error between the predicted variations of BG levels for this participant and the real values caused by the metabolic dynamics for that participant are tried to be minimized. When transferring the model trained for one participant to a different one, the average RMSE values that have been obtained in this scenario have been 49.38 mg/dL which indicate that the model cannot be directly transferred to new users. As a future study, a bigger dataset will be used in order to train the model with the information of a significant variety of participants in order to assess its transferability to other participants.

### 5.3. Comparing Results with Previous Related Studies

The results for RMSE figures in mg/dL for previous related studies are captured in Table 4. Although different datasets are used for the studies, some using real users and some using different diabetes simulators to generate data, the results achieved by the algorithm proposed in this paper show very promising numbers. The optimal results for the model in this paper are achieved for simulated data which could be expected since their deterministic approach to data generation models and since the input signals used in the simulator are the same as the ones used for training and validating the model. For a real use case, the glucose dynamics models are more complex and are influenced by other factors such as physical activity or mental stress.

Most of the models in Table 4 are purely based on the glucose readings from a GCM device which lacks important information from other inputs such as meals and insulin boluses. The best published results that have been found for glucose predictions using a real patient dataset are presented in [38], where 15 T1DM patients following a multiple dose insulin therapy were monitored from 5 to 22 days in free-living conditions. The authors make use of meals and insulin data as inputs, but they also add physical activity as an additional input which is not considered in the AIDA dataset used in this paper. The results for real user data in our case show similar RMSE values as those in [38] in a sub-optimal setting (not taking physical activity into account in the model).

## 6. Conclusions

The results captured in Table 4 show that some patterns controlling the evolution of the BG level signal for T1DM patients can be learnt by using different machine learning techniques. Depending on the input signals and the machine learning methods, different accuracy figures are achieved when trying to predict upcoming values for the BG signal. Table 4 also shows that implementing more complex machine learning models does not necessarily mean achieving better results. In fact, the best results in previous studies captured in Table 4 are achieved by a support vector machine used for regression purposes, known as Support Vector Regression (SVR), when some of the features used as inputs are derived from generic metabolic models for insulin and carbohydrate absorption. This paper proposes, implements, validates and compares a new hybrid model that imbricates the differential equations in metabolic models inside a deep machine learning structure in order to mimic the metabolic behavior of physiological blood glucose models and be trainable for each patient.

The model works better for simulated patients since the complexity of the dependencies from insulin and carbohydrate intake in BG levels are limited to a control set of configuration parameters. Using the AIDA diabetes simulator [4,5] an RMSE of 3.45 mg/dL is achieved for a 30 min prediction horizon when using a 70–30% random data split for training and validation. Different configurations for the size of memory cells inside the proposed model have been tested, validating that there is an optimal value for the complexity of the model (a more complex model does not necessarily achieve better results, but a model adapted to the internal physiological dependencies among the input signals and the body metabolism). In our case, the optimal value for the LSTM memory units was 10.

The model is also able to learn from real patients. Using the dataset in [6], with nine real T1DM patients the model achieves results under 10 mg/dL for the prediction horizon on 30 min when trained for each particular patient/participant using part of the data in the dataset and validating the same patients with the rest of the data (using again a 70–30% random data split for training and validation). The model trained for one user does not necessarily achieve good results for predicting upcoming glucose levels for other patients/participants (in fact the glucose dynamics models use tunable parameters for each particular user).

One limitation of the dataset in [6] is the number of days in which the data is recorded for each participant. As a future work, the model will be used to predict upcoming values for BG levels using other datasets.

## Figures and Tables

**Figure 1 sensors-20-03896-f001:**
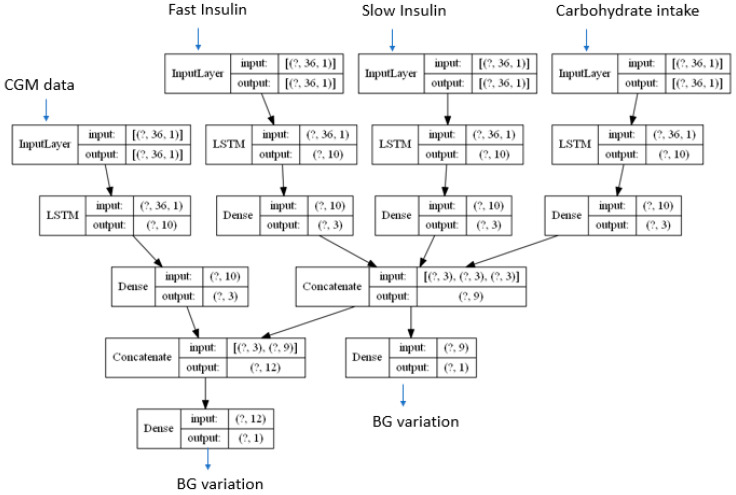
Proposed model.

**Figure 2 sensors-20-03896-f002:**
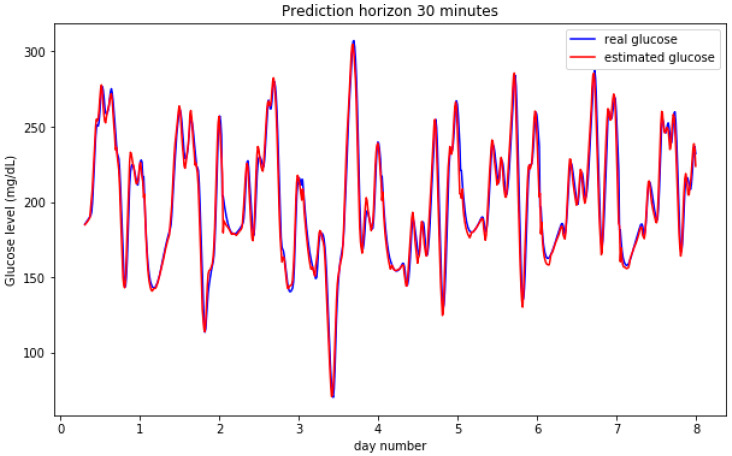
Predictions on a 30 min horizon for 8 days for a simulated participant.

**Figure 3 sensors-20-03896-f003:**
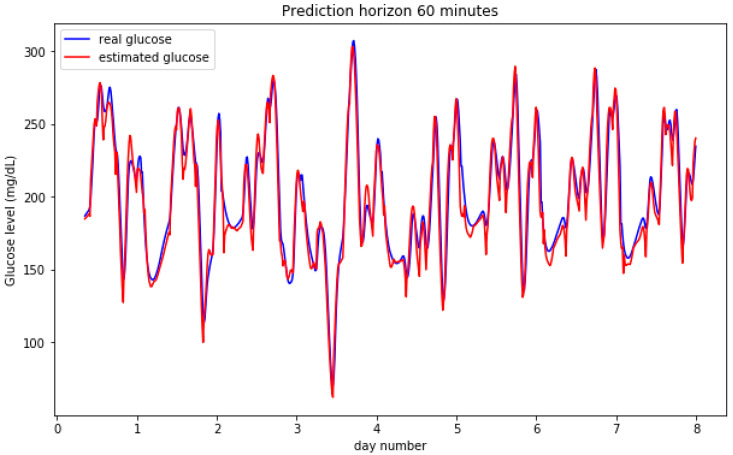
Predictions on a 60 min horizon for 8 days for a simulated participant.

**Figure 4 sensors-20-03896-f004:**
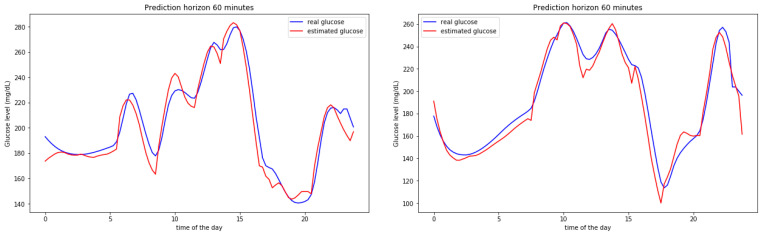
Predictions on a 60 min horizon for 2 single days for a simulated participant.

**Figure 5 sensors-20-03896-f005:**
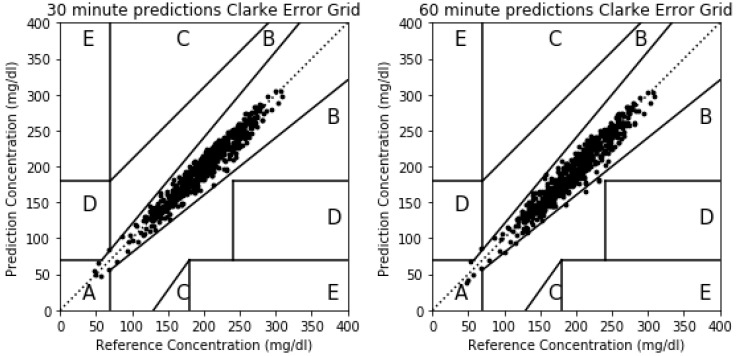
Clarke Error Grid for predictions on a 30 and 60 min horizon for a simulated participant.

**Figure 6 sensors-20-03896-f006:**
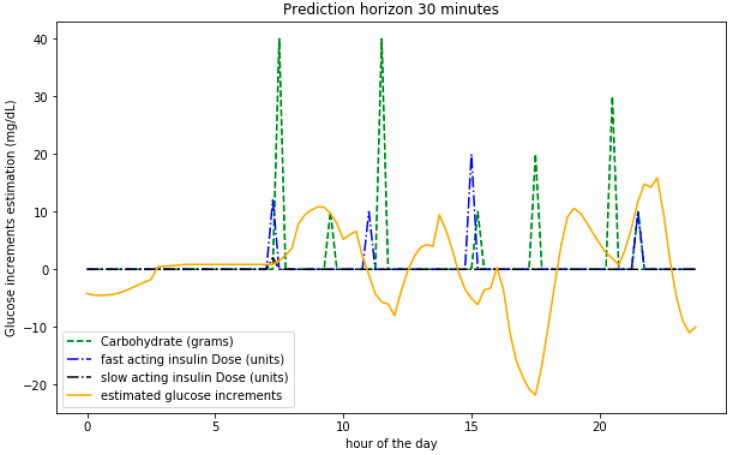
Predictions for BG variations on a 30 min horizon for a single day for a participant based on insulin and meals.

**Figure 7 sensors-20-03896-f007:**
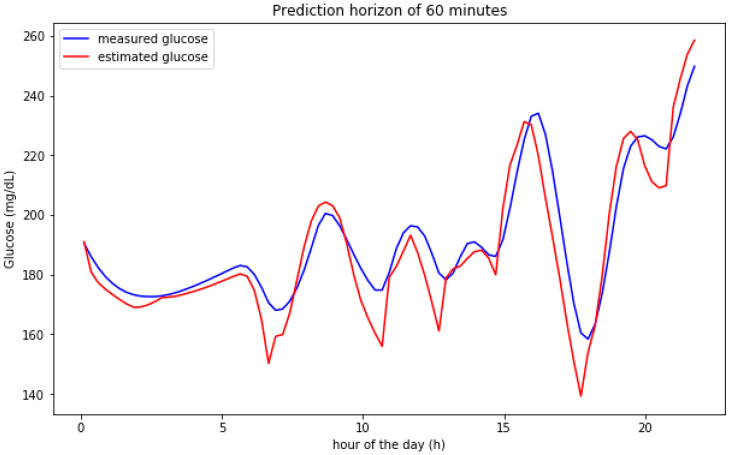
Predictions on a 60 min horizon for a single day for a simulated participant.

**Figure 8 sensors-20-03896-f008:**
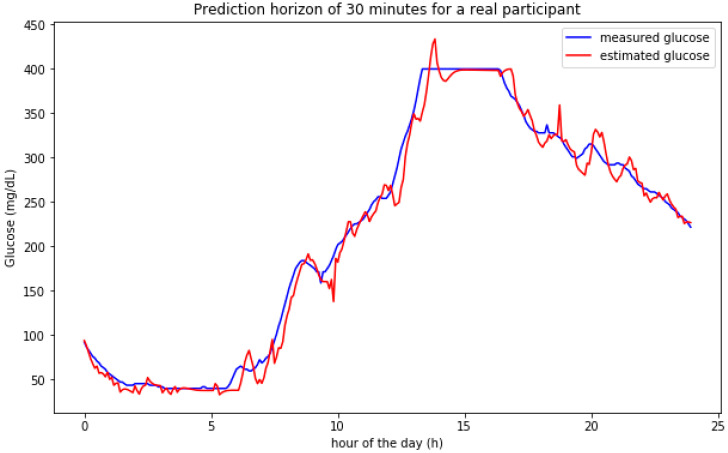
Predictions on a 30 min horizon for a single day for a real participant.

**Figure 9 sensors-20-03896-f009:**
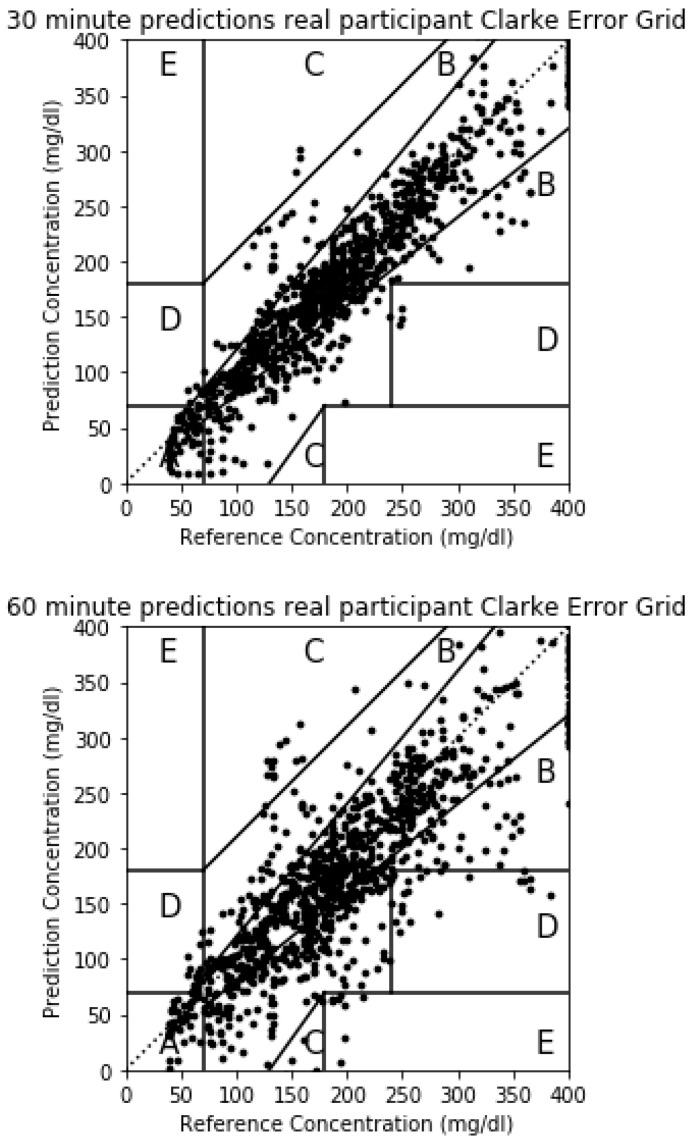
The 30 and 60 min horizon Clarke Error Grids for a real participant.

**Table 1 sensors-20-03896-t001:** Root Mean Square Error (RMSE) values in mg/dL. In total, 70% training 30% validation.

Number of Memory Cells	30 min	60 min
5	3.53	7.09
10	3.45	4.72
15	3.47	6.45

**Table 2 sensors-20-03896-t002:** RMSE values in mg/dL. A total of 8 days of data for training with 2 different days for validation.

Number of Memory Cells	30 min	60 min
5	2.77	7.1
10	2.83	4.35
15	2.63	5.76

**Table 3 sensors-20-03896-t003:** RMSE values in mg/dL split70% for training and 30% for validation.

Number of Memory Cells	30 min	60 min
10	6.42	11.35

**Table 4 sensors-20-03896-t004:** RMSE for 30 min Blood Glucose (BG) estimation using deep (underlined) and shallow learning models.

Study	Input Variables	Method Used	RMSE (mg/dL)
Li et al. [24]	GCM data	Echo State Network (ESN)	23.57
Zhu et al. [28]	CGM data, insulin and carbohydrate	Causal CNN	21.7
Sun et al. [22]	GCM data	RNN-LSTM	21.7
Martinsson et al. [20]	GCM data	RNN-LSTM	20.1
Sparacino et al. [39]	CGM Data	AR	18.78
Pérez-Gandia et al. [40]	CGM Data	Feed-Forward NN	17.5
Zecchin et al. [41]	CGM data, glucose rate after meals	Feed-Forward NN and first-order polynomial model	14.0
Idriss [23]	GCM data	RNN-LSTM	12.38
Turksoy et al. [42]	CGM data, insulin on board, energy expenditure, galvanic skin response	Recursive ARMAX model	11.7
Hamdi et al. [15]	CGM data	SVR and DE	10.78
Li et al. [30]	CGM data, insulin and carbohydrate	CNN+RNN-LSTM	9.38
Mosquera-Lopez et al. [29]	CGM and insulin	RNN-LSTM	7.55
Ali et al. [16]	CGM Data	Feed-Forward NN	7.45
**Our model for real patients**	**CGM data, insulin and carbohydrate**	**Metabolic inspired model using RNN-LSTM**	**6.42 ^1^**
Georga et al. [38]	CGM data, meal intake, insulin concentration, energy expenditure, time	SVR—Random Forest (RF)	5.7
**Our model for simulated patients**	**CGM data, insulin and carbohydrate**	**Metabolic inspired model using RNN-LSTM**	**3.45 ^1^**

^1^ For a 70% training 30% validation split.

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
