# Peer review of "Deep Physiological Model for Blood Glucose Prediction in T1DM Patients"

_sensors, 2020, doi:10.3390/s20143896_

Round 1

Reviewer 1 Report

This manuscript provides an overview of approaches to the problem of forecasting blood glucose (BG) evolution in T1DM-patients. In fact, such overviews are not so seldom in the literature (see, for example, the book "Prediction Methods for blood glucose concentration. Design, Use and Evaluation", Springer, 2015), but the considered manuscript reviews some quite recent articles, and that may justify its publication.

At the same time, the author completely misses an important type of blood glucose predictors that are based on the so-called meta-learning algorithms. Note that such predictors are started to be included in recent surveys (see
the above mentioned book, as as well as just published review "Data-driven modeling and prediction of blood glucose dynamics: Machine learning applications in type 1 diabetes" in Artificial Intelligence in Medicine
Volume 98, July 2019, Pages 109-134). They have an important feature to be portable from patient to patient without any adjustments while outperforming other algorithms in terms of clinical accuracy. This feature opens the way for
using them in diabetes smartphone applications. The corresponding patent has been recently granted in several countries (see, e.g., https://patents.google.com/patent/US10307109B2/en). Therefore, I think that for the
sake of completeness meta-learning based predictors should be included in the overview.

The manuscript also describes a new architecture of BG-predictors, called hybrid model, that is based on artificial neural networks, more precisely, on Recurrent Neural Networks, and attempts to incorporate some features of physiological BG-models. The performance of hybrid model predictors is estimated in terms of the root mean square error (RMSE). Such error metric is widely used in the literature, although it is proven to be inadequate for measuring clinical accuracy, especially in hypoglycemic range. Therefore, some other metrics such as Clarke Error Grid Analysis (EGA), Continuous Glucose Error Grid Analysis (CG-EGA) and Prediction Error Grid Analysis (PRED-EGA) have been suggested to quantify the accuracy of BG-reading and prediction. More detailed discussion can be found, for example, in Diabetes Technol. Ther., 2011 Aug;13(8):787-96. The author is encouraged to present the assessment of the proposed hybrid model predictors in the above metrics, or at least indicate that such assessment should accompany the one in terms RMSE.

This reviewer will support a publication of the considered manuscript provided the above issues are properly taken into account.

Author Response

I would like to thank all the reviewers for their positive feedback and constructive suggestions that have contributed to improve the quality of the paper. All the comments have been fully and thoroughly taken into consideration and all the requested changes have been implemented in the revised paper. This document contains details about the changes applied to the manuscript.

Reviewer 1

Comment:

This manuscript provides an overview of approaches to the problem of forecasting blood glucose (BG) evolution in T1DM-patients. In fact, such overviews are not so seldom in the literature (see, for example, the book "Prediction Methods for blood glucose concentration. Design, Use and Evaluation", Springer, 2015), but the considered manuscript reviews some quite recent articles, and that may justify its publication.

Reply:

We thank the reviewer for his/her comment. As stated, there are some previous literature reviews that have presented previous research on T1DM prediction models. The first contribution of the paper is adding major recent contributions to provide an update to previous studies.

Comment:

At the same time, the author completely misses an important type of blood glucose predictors that are based on the so-called meta-learning algorithms. Note that such predictors are started to be included in recent surveys (see the above mentioned book, as as well as just published review "Data-driven modeling and prediction of blood glucose dynamics: Machine learning applications in type 1 diabetes" in Artificial Intelligence in Medicine Volume 98, July 2019, Pages 109-134). They have an important feature to be portable from patient to patient without any adjustments while outperforming other algorithms in terms of clinical accuracy. This feature opens the way for using them in diabetes smartphone applications. The corresponding patent has been recently granted in several countries (see, e.g., https://patents.google.com/patent/US10307109B2/en). Therefore, I think that for the sake of completeness meta-learning based predictors should be included in the overview.

Reply:

A new reference has been added to the paper in order to make the study more complete by incorporating meta-learning modeling:

  1. Naumova, V., Pereverzyev, S. V., & Sivananthan, S. (2012). A meta-learning approach to the regularized learning—Case study: Blood glucose prediction. Neural Networks, 33, 181-193.

The main text (Section 2. Related work) has also been extended with the following information:

The authors in [41] proposed a meta-learning approach based on the idea of using regularized learning algorithms in predicting blood glucose. Meta-learning approaches are designed to be portable from patient to patient while outperforming other algorithms in terms of clinical accuracy. This feature opens the way for using them in diabetes smartphone applications.

Comment:

The manuscript also describes a new architecture of BG-predictors, called hybrid model, that is based on artificial neural networks, more precisely, on Recurrent Neural Networks, and attempts to incorporate some features of physiological BG-models. The performance of hybrid model predictors is estimated in terms of the root mean square error (RMSE). Such error metric is widely used in the literature, although it is proven to be inadequate for measuring clinical accuracy, especially in hypoglycemic range. Therefore, some other metrics such as Clarke Error Grid Analysis (EGA), Continuous Glucose Error Grid Analysis (CG-EGA) and Prediction Error Grid Analysis (PRED-EGA) have been suggested to quantify the accuracy of BG-reading and prediction. More detailed discussion can be found, for example, in Diabetes Technol. Ther., 2011 Aug;13(8):787-96. The author is encouraged to present the assessment of the proposed hybrid model predictors in the above metrics, or at least indicate that such assessment should accompany the one in terms RMSE.

Reply:

We thank the reviewer for this contribution since adding Clarke Error Grid Analysis is a very good and relevant improvement for showing the results.

The following text has been added to section 5:

Although RMSE has been widely used in previous publications and is therefore a convenient metric for comparing results among them, the Clarke Error Grid Analysis (EGA) is preferred in terms of clinical accuracy of blood glucose estimates [42]. The Clarke Error Grid captures the differences between the estimations for blood glucose levels over the selected horizon and the real measurements for the same instants of time. The Clarke Error Grid divides the bidimensional space into 5 different zones (A to E), each of them showing different implications in terms of blood glucose management. Zone A includes de predicted values that differ from the real values no more than 20 percent or the values in the hypoglycemic range (<70 mg/dl). The points in this zone are considered clinically accurate. Points in Zone B capture values in which the difference between the prediction and real measurements are bigger than 20 percent but would lead to benign or no treatment scenarios. This zone is clinically acceptable. Zone C leads to estimations that imply overcorrecting behaviors based on the prediction errors for blood glucose levels. Points in Zone D represent a failure to detect and treat deviations in blood glucose levels in which the actual levels are outside of the acceptable levels while the predictions fall within the acceptable range. Finally, Zone E captures points in which predicted values are opposite to real/measured blood glucose levels, and therefore, the treatment would go in the opposite direction to what is recommended. In this paper, RMSE values will be used to compare results with previous studies and the Clarke Error Grid will also be used to asses the severity or the errors in the proposed methods in clinical terms.

A new reference is added:

  1. Clarke, W. L., Cox, D., Gonder-Frederick, L. A., Carter, W., & Pohl, S. L. (1987). Evaluating clinical accuracy of systems for self-monitoring of blood glucose. Diabetes care, 10(5), 622-628.

Moreover, the following results have been generated for the simulated case:

Figure 5 captures the results for the Clarke Error Grid for a simulated participant for 30 and 60-minute prediction horizons. The majority of the estimations fall in zone A (clinically accurate zone) for both cases. In the 30-minute horizon prediction, from the 739 points in the validation set, 738 fall in zone A and only 1 in zone D. In the 60-minute horizon case, 729 fall in zone A, 5 in zone B and only 1 in zone D. Zone B is clinically acceptable while zone D represents a failure to detect a hyppo or hypperglycemic point in advance.

(see attached file for the figure)

Figure 5. Clarke Error Grid for predictions on a 30 and 60-minute horizon a simulated participant.

And for the real participant case:

The Clarke Error Grid for a real participant is shown in Figure 9. In this case, the prediction errors captured in Table 3 are bigger than the simulated case and the points outside the Zone A in the Clarke Error Grid will be bigger. For the 30-minute horizon, 868 points fall in Zone A, 205 in zone B, 8 in zone C, 13 in zone D and there is no point in zone E. The majority of the points fall therefore in zones A and B which are consider clinically accurate or acceptable. The 8 points in zone C may lead to overreaction and the 13 points in zone D will imply a failure to detect a glycemic episode 30 minutes in advance. For the 60-minute horizone, 582 points are in zone A, 367 in zone B, 77 in zone C, 57 in zone D and 5 in zone E. The majority of the points continue to fall in zones A and B but the number of error in zones C, D and E increase.

(see attached file for the figure)

Figure 9. 30- and 60-minute horizon Clarke Error Grids for a real participant.

Comment:

This reviewer will support a publication of the considered manuscript provided the above issues are properly taken into account.

Reply:

Thank you very very much for the comments that have been fully implemented as described above.

Reviewer 2 Report

This paper surveyed the state of the art in blood glucose predictions for T1DM patients and proposes, implements, validates and compares a new hybrid model that decomposes a deep machine learning model. Authors proposed A specific Long Short Term Memory based Recurrent Neural Network (LSTM RNN) is used to learn the carbohydrate digestion and insulin
absorption processes from each input signal. The individual effect for each digestion and absorption process after the LSTM RNN layer is combined in order to assess the blood glucose variations for the next Continuous Glucose Monitor (CGM) reading. For improving this manuscript, I have the following comments:

  • In abstract, it would be better to give more detailed content or process of the proposed scheme.  In this version, much portion was covered by introduction like it.
  • Eq. (1) is very low quality. It should be improved.
  • In the proposed model, why we need two BG variations as output?  Authors would be better to explain this reason.
  • In real situation, we can see some large error. This can give some effect on the patients with large amount of error.  I think it should be analyzed its effect in medical aspect.
  • There are some deep learning schemes for medical usage such as emotion recognition. Authors can add the some works as: 

         . Deep Joint Spatio-Temporal Network (DJSTN) for Efficient Facial Expression RecognitionSensors (MDPI), 2020, 20, 1936, doi:10.3390/s20071936, pp. 1-23, March 30, 2020.

         . A Review of Personalized Blood Glucose Prediction Strategies for T1DM Patients, Int J Numer Method Biomed Eng. 2017 Jun;33(6).

        . Blood glucose prediction with variance estimation using recurrent neural networks, Journal of Healthcare Informatics Research 2020, 4, 1–18.

      - There are some typos. Authors should check on the whole manuscript.

I recommend this to be revised for re-consideration in this journal.

Author Response

(see attached file for figure and equation)

I would like to thank all the reviewers for their positive feedback and constructive suggestions that have contributed to improve the quality of the paper. All the comments have been fully and thoroughly taken into consideration and all the requested changes have been implemented in the revised paper. This document contains details about the changes applied to the manuscript.

Reviewer 2

This paper surveyed the state of the art in blood glucose predictions for T1DM patients and proposes, implements, validates and compares a new hybrid model that decomposes a deep machine learning model. Authors proposed A specific Long Short Term Memory based Recurrent Neural Network (LSTM RNN) is used to learn the carbohydrate digestion and insulin
absorption processes from each input signal. The individual effect for each digestion and absorption process after the LSTM RNN layer is combined in order to assess the blood glucose variations for the next Continuous Glucose Monitor (CGM) reading. For improving this manuscript, I have the following comments:

Comment:

In abstract, it would be better to give more detailed content or process of the proposed scheme.  In this version, much portion was covered by introduction like it.

Reply:

We thank the reviewer for his/her comment which has allowed us to show that more detail could be added to the abstract (respecting the word limit for it). We have added the following text to the abstract:

The differential equations for carbohydrate and insulin abortion in physiological models are modeled using a Recurrent Neural Network (RNN) implemented using Long Short Term Memory (LSTM) cells.

Comment:

Eq. (1) is very low quality. It should be improved.

Reply:

Eq. (1) has been improved:

Comment:

In the proposed model, why we need two BG variations as output?  Authors would be better to explain this reason.

Reply:

This is based on the differential equations describing the blood glucose metabolic process which provide values for the increment of blood glucose levels [14][30-31]. The following text has been added to the paper:

The model will learn the blood glucose (BG) dynamics estimating the variation for the next expected CGM measure as captured from the differential equations in metabolic models [14][30-31].

Comment:

In real situation, we can see some large error. This can give some effect on the patients with large amount of error.  I think it should be analyzed its effect in medical aspect.

Reply:

The reviewer is right. RMSE figures are convenient for comparing results with previous research studies but the Clarke Error Grid Analysis provides a better interpretation of results in terms of medical aspects.

The following text has been added to section 5:

Although RMSE has been widely used in previous publications and is therefore a convenient metric for comparing results among them, the Clarke Error Grid Analysis (EGA) is preferred in terms of clinical accuracy of blood glucose estimates [42]. The Clarke Error Grid captures the differences between the estimations for blood glucose levels over the selected horizon and the real measurements for the same instants of time. The Clarke Error Grid divides the bidimensional space into 5 different zones (A to E), each of them showing different implications in terms of blood glucose management. Zone A includes de predicted values that differ from the real values no more than 20 percent or the values in the hypoglycemic range (<70 mg/dl). The points in this zone are considered clinically accurate. Points in Zone B capture values in which the difference between the prediction and real measurements are bigger than 20 percent but would lead to benign or no treatment scenarios. This zone is clinically acceptable. Zone C leads to estimations that imply overcorrecting behaviors based on the prediction errors for blood glucose levels. Points in Zone D represent a failure to detect and treat deviations in blood glucose levels in which the actual levels are outside of the acceptable levels while the predictions fall within the acceptable range. Finally, Zone E captures points in which predicted values are opposite to real/measured blood glucose levels, and therefore, the treatment would go in the opposite direction to what is recommended. In this paper, RMSE values will be used to compare results with previous studies and the Clarke Error Grid will also be used to asses the severity or the errors in the proposed methods in clinical terms.

A new reference is added:

  1. Clarke, W. L., Cox, D., Gonder-Frederick, L. A., Carter, W., & Pohl, S. L. (1987). Evaluating clinical accuracy of systems for self-monitoring of blood glucose. Diabetes care, 10(5), 622-628.

Moreover, the following results have been generated for the simulated case:

Figure 5 captures the results for the Clarke Error Grid for a simulated participant for 30 and 60-minute prediction horizons. The majority of the estimations fall in zone A (clinically accurate zone) for both cases. In the 30-minute horizon prediction, from the 739 points in the validation set, 738 fall in zone A and only 1 in zone D. In the 60-minute horizon case, 729 fall in zone A, 5 in zone B and only 1 in zone D. Zone B is clinically acceptable while zone D represents a failure to detect a hyppo or hypperglycemic point in advance.

Figure 5. Clarke Error Grid for predictions on a 30 and 60-minute horizon a simulated participant.

And for the real participant case:

The Clarke Error Grid for a real participant is shown in Figure 9. In this case, the prediction errors captured in Table 3 are bigger than the simulated case and the points outside the Zone A in the Clarke Error Grid will be bigger. For the 30-minute horizon, 868 points fall in Zone A, 205 in zone B, 8 in zone C, 13 in zone D and there is no point in zone E. The majority of the points fall therefore in zones A and B which are consider clinically accurate or acceptable. The 8 points in zone C may lead to overreaction and the 13 points in zone D will imply a failure to detect a glycemic episode 30 minutes in advance. For the 60-minute horizone, 582 points are in zone A, 367 in zone B, 77 in zone C, 57 in zone D and 5 in zone E. The majority of the points continue to fall in zones A and B but the number of error in zones C, D and E increase.

Figure 9. 30- and 60-minute horizon Clarke Error Grids for a real participant.

Comment:

There are some deep learning schemes for medical usage such as emotion recognition. Authors can add the some works as: 

         . Deep Joint Spatio-Temporal Network (DJSTN) for Efficient Facial Expression Recognition, Sensors (MDPI), 2020, 20, 1936, doi:10.3390/s20071936, pp. 1-23, March 30, 2020.

         . A Review of Personalized Blood Glucose Prediction Strategies for T1DM Patients, Int J Numer Method Biomed Eng. 2017 Jun;33(6).

        . Blood glucose prediction with variance estimation using recurrent neural networks, Journal of Healthcare Informatics Research 2020, 4, 1–18.

 Reply:

We thank the reviewer for his/her contribution with these very much related references. The manuscript has been enhanced by incorporating the following references:

  • Oviedo S., Vehi J., Calm R. and Armengol J. A Review of Personalized Blood Glucose Prediction Strategies for T1DM Patients, Int J Numer Method Biomed Eng. 2017 Jun;33(6).
  • Martinsson, J., Schliep, A., Eliasson, B., & Mogren, O. (2020). Blood glucose prediction with variance estimation using recurrent neural networks. Journal of Healthcare Informatics Research, 4(1), 1-18.

The following text has also been added:

A review comparing physiological models partially or totally replaced by machine learning techniques can be found in [43]. 

The authors in [44] proposed an approach based on recurrent neural networks (RNN) trained in an end-to-end fashion, using the blood glucose signal, able to provide an estimate of the certainty in the predictions by training the recurrent neural network to parameterize a univariate Gaussian distribution over the output.

Comment:

- There are some typos. Authors should check on the whole manuscript.

Reply:

The paper has been fully revised as proposed.

Comment:

I recommend this to be revised for re-consideration in this journal

 Reply:

We thank the reviewer for his/her positive comments that have allowed us to improve the paper.

Reviewer 3 Report

The present ms proposes a prediction model for blood glucose. The ms is well written, and I have only a few minor comments:

  1. line 256ff.: More implementation details would help replicate the findings- It is not sufficient to state that the „model […] has been implemented in Python using Keras and Tensorflow“. More details about used functions or code snippets could be included in an appendix or supplementary material.
  2. When referring to sections, tables, or figures in the ms, the upper case has to be used, e.g. (line 189) „… is captured in Figure 1“.
  3. The reference list has some inconsistencies, e.g., lower and upper case notation in journal titles or inconsistent use of journal abbreviations.

Author Response

I would like to thank all the reviewers for their positive feedback and constructive suggestions that have contributed to improve the quality of the paper. All the comments have been fully and thoroughly taken into consideration and all the requested changes have been implemented in the revised paper. This document contains details about the changes applied to the manuscript.

Reviewer 3

The present ms proposes a prediction model for blood glucose. The ms is well written, and I have only a few minor comments:

Comment:

line 256ff.: More implementation details would help replicate the findings- It is not sufficient to state that the „model […] has been implemented in Python using Keras and Tensorflow“. More details about used functions or code snippets could be included in an appendix or supplementary material.

Reply:

We thank the reviewer for his/her comment and a new appendix has been added with the Python code implementing the model so that the results can be reproduced.

The content added to the manuscript is:

Appendix A. Model implementation

This appendix shows the model implementation in Python so that results can be replicated.

if (trainable):

    input1 = Input(shape=(time_span, 1))

    x11 = LSTM(units=mem_cells, activation='relu', return_sequences=False)

    x12 = x11(input1)

    x13 = Dense(units=3, activation='relu')

    x1 = x13(x12)

    input2 = Input(shape=(time_span,1))

    x21 = LSTM(units=mem_cells, activation='relu', return_sequences=False)

    x22 = x21(input2)

    x23 = Dense(units=3, activation='relu')

    x2 = x23(x22)

    input3 = Input(shape=(time_span, 1))

    x31 = LSTM(units=mem_cells, activation='relu', return_sequences=False)

    x32 = x31(input3)

    x33 = Dense(units=3, activation='relu')

    x3 = x33(x32)

    input4= Input(shape=(time_span,1))

    x41 = LSTM(units=mem_cells, activation='relu', return_sequences=False)

    x42 = x41(input4)

    x43 = Dense(units=3, activation='relu')

    x4 = x43(x42)

    added = Concatenate(axis=-1)([x2, x3, x4])

    out1 = Dense(1)(added)

    added2 = Concatenate(axis=-1)([x1, added])

    out2 = Dense(1)(added2)

    model = Model(inputs=[input1, input2, input3, input4], outputs=[out1, out2])

    model.compile(loss='mean_squared_error', optimizer=keras.optimizers.Adam(0.001))

    history = model.fit([xTrain[:,:,4:5],xTrain[:,:,1:2],xTrain[:,:,2:3],xTrain[:,:,3:4]], [yTrain, yTrain],

        epochs=100,

        batch_size=bs,

        validation_split=0.3,

        verbose=1,

        shuffle=False)

Comment:

When referring to sections, tables, or figures in the ms, the upper case has to be used, e.g. (line 189) „… is captured in Figure 1“.

Reply:

All references to figures have been uppercased.

Comment:

The reference list has some inconsistencies, e.g., lower and upper case notation in journal titles or inconsistent use of journal abbreviations.

Reply:

The references have been reviewed.

Reviewer 4 Report

This paper provided a review of the methods of blood glucose predictions for T1DM patients. And the author also compares a deep machine learning model in order to mimic the metabolic behavior of physiological blood glucose methods. The first is about the comparison studies. As pointed out, numerous methods have been available for this prediction. The comparison study need be organized according to the categories. And the comparison need be based on a single standard. This is very important for showing the advantages/characteristics of the comparing methods. Moreover, the applications of this method need be discussed. How to use this method in real conditions for biomedical instruments. Some typos need be fixed before publication. Some abbreviations need to show their original words at the first time, such as RMSE in the abstract. Thanks.

Author Response

(see attached file)

I would like to thank all the reviewers for their positive feedback and constructive suggestions that have contributed to improve the quality of the paper. All the comments have been fully and thoroughly taken into consideration and all the requested changes have been implemented in the revised paper. This document contains details about the changes applied to the manuscript.

Reviewer 4

This paper provided a review of the methods of blood glucose predictions for T1DM patients. And the author also compares a deep machine learning model in order to mimic the metabolic behavior of physiological blood glucose methods.

Comment:

The first is about the comparison studies. As pointed out, numerous methods have been available for this prediction. The comparison study need be organized according to the categories. And the comparison need be based on a single standard. This is very important for showing the advantages/characteristics of the comparing methods.

Reply:

Thank you very much for the suggestion.

We have divided the machine learning models used in previous studies into the two major categories according to the suggestion: shallow and deep learning models and capture the results in table 4.

As recommended by the reviewer, it is important to use a common standard to compare results. The common metric in previous studies is based on using the RMSE error in order to compute the accuracy of each model. The paper has also used the RMSE error for the proposed method and the comparison with previous methods is captured in Table 4.

Moreover, in order to provide further clinical related assessment for the results, the following text is added to the paper:

Previous research studies use different metrics in order to assess the quality of the proposed blood glucose models. Among them, the most widely used figure is the Root Mean Square Error (RMSE) defined by the following equation:

                                                       ,                                                     

(1)

where Gm represents the measured BG level and Ge is used for the estimated value for BG level at the same future instant of time.

Although RMSE has been widely used in previous publications and is therefore a convenient metric for comparing results among them, the Clarke Error Grid Analysis (EGA) is preferred in terms of clinical accuracy of blood glucose estimates [42]. The Clarke Error Grid captures the differences between the estimations for blood glucose levels over the selected horizon and the real measurements for the same instants of time. The Clarke Error Grid divides the bidimensional space into 5 different zones (A to E), each of them showing different implications in terms of blood glucose management. Zone A includes de predicted values that differ from the real values no more than 20 percent or the values in the hypoglycemic range (<70 mg/dl). The points in this zone are considered clinically accurate. Points in Zone B capture values in which the difference between the prediction and real measurements are bigger than 20 percent but would lead to benign or no treatment scenarios. This zone is clinically acceptable. Zone C leads to estimations that imply overcorrecting behaviors based on the prediction errors for blood glucose levels. Points in Zone D represent a failure to detect and treat deviations in blood glucose levels in which the actual levels are outside of the acceptable levels while the predictions fall within the acceptable range. Finally, Zone E captures points in which predicted values are opposite to real/measured blood glucose levels, and therefore, the treatment would go in the opposite direction to what is recommended. In this paper, RMSE values will be used to compare results with previous studies and the Clarke Error Grid will also be used to assess the severity or the errors in the proposed methods in clinical terms.

Comment:

Moreover, the applications of this method need be discussed. How to use this method in real conditions for biomedical instruments.

Reply:

Thank you very much for this comment. We fully agree.

The introduction of the paper states that:

Very high or low values of BG levels can cause different inconveniencies and damages to the human body. Mechanisms able to anticipate them could increase the quality and even save lives for those suffering T1DM.

Providing models for accurate prediction of glucose levels in T1DM patients is critical both for their glycemic control and for the development of closed-loop systems [3].

The following information has been added to show the application of the method:

This paper focuses on the use of the most common data sources/signals used in previous research studies for BG level estimation over prediction horizons of 30 to 60 minutes: current and past BG measurements from CGM devices, fast and slow acting insulin injections and food intake. A horizon of 30 to 60 minutes of accurate predictions will allow the patient to modify insulin or meal intakes with enough time for the insulin and carbohydrate absorption in order to prevent adverse glycemic events.

Comment:

Some typos need be fixed before publication. Some abbreviations need to show their original words at the first time, such as RMSE in the abstract. Thanks.

Reply:

A complete review has been carried out. Thank you very much. The abbreviations have also been revised.

Round 2

Reviewer 2 Report

I think this paper has been well revised based on my comments and its quality was improved significantly. I recommend this paper to be accepted with its current form.